# The Use of Parenteral Opioids in Cancer Pain Management

**DOI:** 10.3390/cancers15153778

**Published:** 2023-07-25

**Authors:** Sebastiano Mercadante

**Affiliations:** La Maddalena Cancer Center, Via San Lorenzo 312, 90146 Palermo, Italy; terapiadeldolore@lamaddalenanet.it or 03sebelle@gmail.com

**Keywords:** cancer pain, palliative care, parenteral opioids, intravenous morphine

## Abstract

**Simple Summary:**

There are some clinical circumstances that require the parenteral administration of opioids, as an alternative to the oral route. A parenteral route of administration is often indicated in patients with severe pain intensity requiring rapid analgesia or in those who develop nausea and vomiting, dysphagia, or have limited intestinal surface for drug absorption. Subcutaneous and intravenous routes are effective in providing rapid analgesia, particularly the intravenous route. Many opioids have proved to be effective when administered in clinical conditions where oral opioids cannot be used or are poorly effective. Although most studies on this topic are not controlled, there is a body of experience suggesting that parenteral opioids may be useful in some specific circumstances.

**Abstract:**

Opioids should be offered to patients with moderate-to-severe pain related to cancer or active cancer treatment unless contraindicated. Although oral administration of opioids is generally preferable, a parenteral route may be advisable and mandatory in some clinical circumstances. Parenteral administration of opioids may accelerate the achievement of analgesia. The intravenous route fits the need of rapid achievement of analgesia in patients poorly responsive to other opioids and provides a fast analgesia in patients with breakthrough pain, that has a specific temporal pattern requiring a rapid analgesic effect. When the oral route is unavailable for the presence of nausea, vomiting, or dysphagia. the parenteral route is one of the principal options. Opioids have different pharmacokinetic and pharmacodynamic characteristics and should be chosen according to the individual needs. Thus, the knowledge and experience with these routes of administration are mandatory for anesthesiologists committed to cancer pain management.

## 1. Introduction

Pain is a significant problem in patients with advanced cancer. Despite increased attention to cancer pain, the number of patients with cancer has not significantly changed over the last decades. It has been reported that more than one third of patients receiving anticancer therapy suffer from moderate to severe pain and approximately 50% of those with an advanced cancer experience moderate to severe pain [1]. The lack of improvement is likely due to barriers for patients to report pain, poor pain assessment, limited knowledge of available analgesic treatments, no significant changes in the treatment of neuropathic pain, and limited availability of pharmacological treatment [2].

Cancer pain is challenging and requires high level clinical competence in all available methods to relieve it. World Health Organization (WHO) guidelines for cancer pain management were based on the use of a sequence of analgesic drugs with increasing efficacy to be used according to the level of pain intensity [3]. The aim of this approach was to promote the prescription of opioids, as the principal barrier against appropriate cancer pain management was the reluctance of health care professionals, institutions, and governments to use opioids for fear of addiction, tolerance, and misuse. These guidelines were of paramount importance, as their application provided satisfactory pain relief in most patients with cancer pain. Despite the extensive experience suggesting the efficacy of this approach, data in terms of evidence-based medicine were not solid. This approach was simplistic and has been erroneously considered the gold standard of cancer pain management that one can use successfully in any case [2].

The wide variety and ranges of cancer pain cannot be included in the three step ladder, and any specific clinical circumstance requires profound knowledge of all available pharmacological techniques. Opioids still remain the cornerstone of cancer pain management and should be prescribed to patients with moderate-to-severe pain related to cancer [4]. The pharmacological management of cancer pain allows acceptable pain control in the majority of patients. An optimization of opioid therapy may allow an individual treatment according to the patients’ characteristics and pain syndromes, providing timely alternatives in the different stages of disease. However, different phases of disease may require different approaches. Although oral administration of opioids is generally preferable, the parenteral route may be advisable in some clinical circumstances that include acute pain exacerbations leading to an emergency room admission, uncontrolled pain due to progression of disease, or a loss of analgesia due to a shift of the dose-response curve, or the development of tolerance or hyperalgesia. In addition, the presence of vomiting, dysphagia, a limited intestinal surface for drug absorption, or states of sub-obstruction, require alternative routes to oral administration [5]. Finally, parenteral opioids have been used to facilitate the transition to “slow” routes of administration, such as the trandermal route. Concerns regarding respiratory depression are one of the most important physician barriers to the use of parenteral opioids in treatment, particularly in the use of opioid dose titration. This narrative review examines the historical background, as well as more recent studies reporting different drugs, modalities, and settings on the parentaeral administration of opioids for cancer pain management.

## 2. Preliminary Historical Background

Subcutaneous and intravenous routes have been reported for the management of cancer pain by early studies. These studies were the basis of subsequent studies published later in the third millennium, which will be examined in detail in the subsequent paragraphs. Opioids may be administered parenterally by either continuous infusion or regularly scheduled intermittent injections. The relative advantages and disadvantages of each modality are not very clear. Continuous infusion produces steady drug levels, thus avoiding the “peak effect” of adverse effects, such as nausea and sedation, immediately after an injection, and breakthrough pain prior to the next dose (i.e., end of dose failure). Its use in the treatment of cancer pain is historically well established [6]. Administration of opioids by intermittent injection is associated with fluctuations in drug levels [7]. Some studies have reported the use of an inexpensive, lightweight, manually operated portable device for intermittent subcutaneous injection of opioid through an indwelling needle. A retrospective analysis of its use in patients with advanced cancer reported an effective analgesia without unacceptable toxicity, with minimal costs, and successful use in the home setting [8]. In a randomized double-blind crossover trial comparing analgesic and adverse outcomes between continuous subcutaneous infusion and regularly scheduled intermittent subcutaneous injections for opioid administration for cancer pain, no significant differences between the two modalities were found [9].

Previous studies reported that approximately 15% of cancer patients were administered opioids by intravenous route prior to death [10]. The first step in defining the indications for a continuous infusion requires the distinction between the intravenous route of administration and continuous infusion, which requires an extra step in preparation and a specialized drug delivery system. The principal indication is when oral and intramuscular routes were not tolerated, for example, in severely cachectic patients who develop bowel obstruction, in pediatric patients unable to take medications by mouth, or in situations where there are concerns over incomplete intestinal absorption in the patient with severe pain. Continuous infusion should be considered to avoid bolus effects that may occur during repetitive dosing. Bolus effects are defined as the occurence of toxicity, mainly sedation, immediately after a dose. Presumably, continuous infusion is able to obviate this problem by maintaining a more constant plasma drug concentration preventing fluctuating peak levels obtained with repetitive dosing.

Previous recommendations on continuous infusion of opioids suggest some indications for continuous infusion (for example, a bolus effect, ease of nursing, or rapid titration of dose) [11]. The drugs should be carefully chosen, considering prior experience with opioids, efficacy of current analgesics, and pharmacokinetic factors, with a preference for drugs with a short half-life. An appropriate infusion device should be chosen. The current total daily opioid consumption should be converted to parenteral equivalents. If an alternative drug is selected, the current daily quantity of opioid consumed should be converted to parenteral equivalents of the infusion drug, delivering one-half this dose over next 24 h. A loading bolus prior to start the infusion should be given. Vital signs should be monitored for some hours after each loading bolus or increase in infusion rate. The infusion rate should be increased until either analgesia or intolerable side effects occur. If side effects develop and cannot be controlled with adjuvant drugs, an alternative method of analgesia should be taken into consideration. A continuous infusion with a different opioid should be considered. Rapid dose titration is not a strong indication, since the rate of rise of plasma drug concentration may be faster with repetitive intravenous boluses. Although individual dose size can become very large in a drug-tolerant patient with pain, infusion rates can also be very high, up to 500 mg per hour of morphine or more.

## 3. Subcutaneous and Intravenous Opioids

Subcutaneous opioid administration is easy and can be effectively used in settings such as hospices, nursing homes, or home care. However, patients with poor peripheral circulation, coagulation disorders, or generalized edema, or those who develop erythema, soreness, or abscesses are unfit for this route of administration [12]. Studies reported that a continuous subcutaneous infusion produced better pain relief and fewer adverse effects with lower opioid doses, when compared to a previous treatment with oral opioids. Efficacy of the start of parenteral opioids in 100 patients with cancer pain who failed on conventional opioids was evaluated. Parenteral opioids improved the balance between analgesia and adverse effects in patients who failed on conventional opioids, with an important improvement seen in more than 70% of patients, possibly demonstrating that this route is a good alternative to spinal opioids. It was also suggested that a change of route alone is as effective as opioid rotation [13].

The intravenous route may be advantageous in patients who already have an indwelling intravenous line. Most patients in specialized units have a long-term intravenous line to provide hydration and to facilitate therapeutic procedures that require a rapid effect, such as opioid dose titration, administration of rescue doses, and treatment of emergencies, other than for injecting urticant chemotherapic agents [5]. Because of its fast analgesic effect, intravenous morphine fits the temporal pattern of breakthrough pain. In several studies, intravenous morphine, given in doses proportional to the background opioid dosage, has been reported to be safe and effective in most patients experiencing breakthrough pain [5].

The principal advantage of the intravenous route over the subcutaneous one is pharmacokinetic: the direct administration into the circulation provides a fast and predictable effect that is independent of absorption problems (Table 1). However, physicians are often reluctant to use the intravenous route because of the possible risks and poor familiarity, particularly in the palliative setting where the subcutaneous route is more familiar. Despite its invasiveness, intravenous opioids may be useful in some clinical situations in a secondary-care setting. In a comparative study of intravenous and subcutaneous morphine, the intravenous route provided a faster time to achieve adequate analgesia compared to subcutaneous administration, although stable and lasting pain relief was observed after four days in both groups [14].

A protocol for cancer pain emergencies has been proposed. After initial assessment, patients were managed according to a protocol whereby intravenous boluses of opioid are administered with rapid upward titration until an effective analgesic dose was found. The initial dose of intravenous opioid was arbitrarily selected within the range of of 10–20 mg of intravenous morphine and was administered over 15 min. The dose was doubled every half hour until the achievement of acceptable analgesia. All patients obtained satisfactory relief of excruciating pain in approximately 90 min [15]. In a study of patients with advanced cancer experiencing severe and prolonged pain, bedside boluses of morphine (2 mg every 2 min) were administered, until a significant analgesia was achieved. The effective dose administrated intravenously was assumed to provide analgesia for about four hours, and the daily continuous intravenous dose was calculated consequently for the next 24 h. Pain intensity significantly decreased within 10 min, with a mean dose of intravenous morphine of 8.5 mg. After 1–2 days of intravenous infusion, it was possible to convert intravenous morphine to oral morphine (using a ratio of 1:3), maintaining a good analgesia until hospital discharge, which occurred within a mean of 4.6 days. No significant adverse effects were recorded [16]. In Figure 1, there is an example of a successful dose titration with intravenous morphine in opioid-tolerant patients receiving 240–480 mg/day of oral morphine equivalents. Effective analgesia is commonly achieved in 10–15 min. The effective bolus dose is assumed to last approximately 4 h, and the dose is then calculated for 24 h by multipling the effective dose by 6. Continuous reassessment and eventual dose adjustments are warranted. For breakthrough pain episodes, the same effective dose can be administered. Once the continuous intravenous daily dose provides an adequate and stable analgesia, generally occurring in 24–48 h, intravenous morphine is converted to oral morphine by using a ratio of 1:3. This approach is considered to be as effective as safe. The same process by oral route would require several days, resulting in unnecessary suffering.

In a randomized controlled study, intravenous morphine was compared against oral morphine for opioid dose titration [17]. However, opioid dose titration was not fast enough. Patients received an intravenous bolus dose of 1.5 mg of morphine, repeated every 10 min until total pain relief was achieved or adverse effects developed. Of concern, the ratio between the initial intravenous dose and subsequent oral dose was 1:1. Indeed, patients included in the study had low opioid requirements and most of them did not have sufficiently severe pain to suggest the need for an intravenous opioid dose titration. From a pharmacokinetic point of view, small repeated boluses are unlikely to produce a cumulative effect, as they produce a series of overlapping curves providing poor effective drug concentrations. This is likely the reason for the finding of an intravenous–oral conversion ratio of 1:1. This ratio is substantially different from that reported in other widely accepted studies, from which, a ratio of 1:3 has been well established. A prospective observational cohort study has reported that the 1:3 ratio provides both efficacy and safety when converting intravenous morphine to oral morphine. About 80% of patients were successfully switched with a ratio of 1:3. Twenty per cent of patients in that study required an oral morphine dose adjustment after route switch for better pain control or for reduction in adverse effects. Thus, a 1:3 intravenous to oral morphine milligram potency ratio appears correct and practical for most patients over a wide morphine dose range [18].

Other studies have examined other opioids. Hydromorphone given by intravenous route, as a continuous infusion or by patient-controlled analgesia (PCA) was effective in dose titration studies [19]. When using intravenous oxycodone for opioid dose titration in patients with severe pain, a significant change in pain intensity was observed in a few minutes and conversion to oral route occurred after a mean of 2–3 days [20]. No differences in effective doses, patient satisfaction or adverse effects were observed between oxycodone and morphine given by continuous infusion [21].

In another comparison study with intravenous morphine, significant analgesia was faster with oxycodone; however, the pain relief stabilization success rates within 24 h were similar [22].

Methadone is a peculiar agent with a broad spectrum activity. In addition to being a potent μ-opioid agonist and possessing some affinity for the κ- and δ-opioid receptors, it exerts extra-opioid activities, including the decrease in monoamine reuptake and inhibition of presynaptic NMDA receptors [23]. Clinical observations have described the use of high-dose intravenous methadone by PCA and continuous infusion. In many case reports, it was necessary to reduce the initial doses of intravenous methadone used after switching from high doses of opioids [24]. These initial observations underline the need for accurate monitoring. Since four to five half-lives are required to approach steady state plasma concentrations, accumulation of methadone may occur for many days after a dose increase. Thus, doses should be flexibly modified according to the clinical response, after an adequate priming, necessary to rapidly achieve an analgesic effect. The maintenance of high doses without strict surveillance may be dangerous.

Other studies provided promising data regarding the use of intravenous methadone in difficult pain conditions [25]. More recently, intravenous methadone dose titration was used for opioid switching in patients poorly responsive to different types of opioids. Unlike reports of morphine treatment [26], the effective dose was multiplied by three to establish the intravenous daily dose to be given as a continuous infusion. Doses were then changed according to the clinical needs. Once the clinical situation was stabilized, the intravenous dose was converted to oral methadone, by using an initial ratio of 1:1.2. Although this approach should be performed by skilled personnel, it resulted in highly effective and safe control of very difficult pain syndromes.

Transdermal fentanyl is unfit for inducing a rapid analgesia, given its pharmacokinetic profile. The approximate steady state is reached 12–24 h after application of the patch. Plasma concentrations remain relatively constant thereafter for 48–60 h. Indeed, intravenous fentanyl has distribution and redistribution times of 1.7 and 13 min, respectively, a terminal elimination half-life of about 220 min, and a duration of analgesic effect of about 60 min. For this reason, the intravenous route was proposed the facilitate the transition from and to transdermal fentanyl [27,28,29]. The use of lipophilic drugs such as fentanyl would potentially result in a faster titration compared to morphine, as peak plasma levels could be obtained in few minutes. Although administering intravenous fentanyl is a familiar proactive for anesthesiologists, some palliative care physicians are not so familiar with this route of administration. Intravenous fentanyl has been used safely in the emergency room to treat patients who needed fast opioid dose titration to control their pain. Regardless of the previous morphine doses (276 mg/day), titration with intravenous fentanyl, with a mean of 214 µg was successful within approximately 11 min, without relevant adverse effects. The initial bolus was 10% of the previous morphine dose [30]. Interestingly, boluses of 100 µg of intravenous fentanyl were effective for treating breakthrough pain in highly opioid-tolerant patients who were unresponsive to other opioids given as needed [31].

## 4. Parenteral Opioids and Respiratory Depression

Concerns regarding respiratory depression are one of the most important physician barriers against the use of parenteral opioid dose titration. However, parenteral opioid dose titration for relief of severe cancer pain has been found to be not associated with respiratory depression. In a study of patients with severe cancer undergoing parenteral dose titration, end-tidal CO_2_ did not significantly change and O_2_ saturation was maintained at ≥92% [32]. This supports the statement that an appropriate dose titration of parenteral opioids for pain control is not associated with respiratory depression, nor a major concern during titration. This is consistent with experimental findings that showed that pain is a natural antagonist to opioid respiratory depression [33].

## 5. Conclusions

There are a variety of indications for using parenteral opioids. Patients with severe pain and poorly responsive to previous opioid therapy are typically candidates for a personalized, rapid opioid dose titration obtainable with parenteral administration. In most cases, this corresponds to an opioid switching, characterized by a more intensive approach. Morphine is the more commonly used drug. In the presence of an intravenous line, the intravenous route is more advisable than the subcutaneous one. The rapidity of opioid dose titration is dependent on the patient’s level of previous opioid tolerance, the staff experience and ability to assess analgesia, the ability to monitor the development of adverse effects, and the capacity to use all available intravenous opioids. When using intravenous methadone, substantial clinical experience is necessary for the use of appropriate dosing according to any changes in the patient’s clinical response. Of concern is that, while intravenous dose titration has been performed in specialized medical care centers with a high level of experience, expertise in settings such as hospice or home care is lacking. In the hospice or home setting, the subcutaneous route is suggested, if not contraindicated, although an accumulation of experience and expertise would be desirable even in these settings. Education and information on the use of parenteral opioids should be implemented, with studies specifically designed with a clinical perspective.

## Figures and Tables

**Figure 1 cancers-15-03778-f001:**
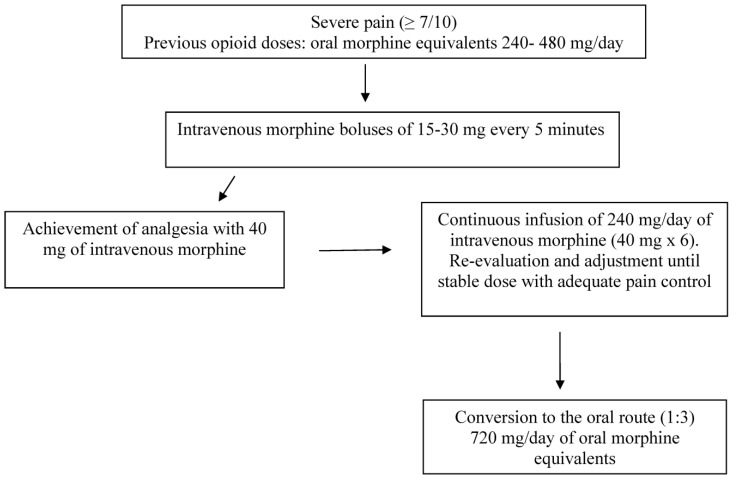
Example of rapid dose titration with intravenous morphine in an opioid-tolerant patient with poor pain control.

**Table 1 cancers-15-03778-t001:** Advantages and disadvantages of intravenous and subcutaneous routes.

	Advantages	Disadvantages
Intravenous route	Rapid effect for dose titration Rapid effects of changes of infusion rate	Need of an intravenous line
Subcutaneous route	Simple (hospice-home setting)	Presence of poor peripheral circulation Edema Soreness Slower titration

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
