# Peer review of "The Use of Parenteral Opioids in Cancer Pain Management"

_cancers, 2023, doi:10.3390/cancers15153778_

Round 1

Reviewer 1 Report

The work entitled "The use of parenteral opioids in cancer pain management" is an interesting narrative mini-review about a subject of very high importance.

Some major comments for consideration before publication:

-Line 92: "Pioneer, but in general still valid, recommendations" - what does the Author mean here? Is pioneer the right word here? Are there no more recent guidelines than the one cited (from 1987)? Also, the word 'pioneer' is used several times throughout the text, but it is not obvious to me what the Author exactly means by that. It would be better to describe the studies more precisely (e.g. prospective/retrospective multicenter clinical study) as currently there is no clear differentiation between the different levels of evidence.

- In line 257, the Author writes: "Parenteral opioid dose titration for relief of severe cancer pain was found to be not associated with respiratory depression"  and cites a small study, with 30 participants from 2007. This part seems to be a bit too simplifying regarding the problem of respiratory depression and should be detailed and argued more. E.g. the PRODIGY trial from 2021 (Journal of Clinical Anesthesia,) found that "among ward patients receiving parenteral opioids, 46% experienced ≥1 respiratory depression episode".

Minor comments:

- The text currently includes only two main chapters: Introduction and Conclusions, therefore should be further structured.

- It was not clear to me whether figure 1. is from or based on citation 16. If so, it should be indicated in the legends. If not, further details should be added regarding the basis of the proposed dosing in fig. 1.

-Line 63: What does the Author mean by the “third century”?

- A table would be a nice addition summarizing the positive and negative sides of parenteral opioid administration (e.g. iv. vs sc.), or the main considerations.

The English language is overall good. There are some typos throughout the text (e.g. lines 57, 62, 88, ) and some phrasing issues (e.g. "in the next future" line 281; or starting sentences with "Of concern, ...", e.g. line 276).

Author Response

Reviewer 1

-Line 92: "Pioneer, but in general still valid, recommendations" - what does the Author mean here? Is pioneer the right word here? Are there no more recent guidelines than the one cited (from 1987)? Also, the word 'pioneer' is used several times throughout the text, but it is not obvious to me what the Author exactly means by that. It would be better to describe the studies more precisely (e.g. prospective/retrospective multicenter clinical study) as currently there is no clear differentiation between the different levels of evidence.

For “pioneer” I intend old previous studies which reported the first data. In this case I’m talking about recommendations rather than studies. I changed some sentences

- In line 257, the Author writes: "Parenteral opioid dose titration for relief of severe cancer pain was found to be not associated with respiratory depression"  and cites a small study, with 30 participants from 2007. This part seems to be a bit too simplifying regarding the problem of respiratory depression and should be detailed and argued more. E.g. the PRODIGY trial from 2021 (Journal of Clinical Anesthesia,) found that "among ward patients receiving parenteral opioids, 46% experienced ≥1 respiratory depression episode".

This study was performed in a postoperative setting where the risk of respiratory depression is higher for obviosu reasons. On the other hand, in this study data were reported not taking into account anesthesia and surgery type. The percentages reported ( 54% and 45% of patients receiving only short-acting opioids or only long-acting opioids experienced ≥1 episode of opioid-induced respiratory depression, respectively) just denote an inexperienced use of opioids for postoperative pain. In our anestheitc department, the frequency of respiratory depression is almost zero, when opioid doses are titrated in the RR. You are right but the context is different.

Minor comments:

- The text currently includes only two main chapters: Introduction and Conclusions, therefore should be further structured.

The principal paragraphs are: Introduction - Preliminary historical background - Subcutaneous and intravenour opioids – Parenteral opioids and respiratory depression - Conclusion

- It was not clear to me whether figure 1. is from or based on citation 16. If so, it should be indicated in the legends. If not, further details should be added regarding the basis of the proposed dosing in fig. 1.

Please see in the text (now in bold) the description. The legend is already on the botton of the figure

-Line 63: What does the Author mean by the “third century”?

Sorry, it is millennium

- A table would be a nice addition summarizing the positive and negative sides of parenteral opioid administration (e.g. iv. vs sc.), or the main considerations.

I added a table

 The English language is overall good. There are some typos throughout the text (e.g. lines 57, 62, 88, ) and some phrasing issues (e.g. "in the next future" line 281; or starting sentences with "Of concern, ...", e.g. line 276).

I removed “in the next future”. “Of concern” seems to be correct

Reviewer 2 Report

You should add some references, especially in the Introduction section, for example, lines 54, 62, 109, 142, 160

The manuscript has 2 sections: Introduction and Conclusion. Please modify accordingly with the guidelines of MDPI journals.

Moderate editing of English language required

Author Response

2 reviewer

You should add some references, especially in the Introduction section, for example, lines 54, 62, 109, 142, 160

All available references were included. Respectfully, not all the sentences require a reference, and could be based on expert opinion

The manuscript has 2 sections: Introduction and Conclusion. Please modify accordingly with the guidelines of MDPI journals.

The principal paragraphs are: Introduction - Preliminary historical background - Subcutaneous and intravenour opioids – Parenteral opioids and respiratory depression - Conclusion